# Papillary Thyroid Microcarcinoma in Thyroid Surgical Practice: Incidental vs. Non-Incidental: A Ten-Year Comparative Study

**DOI:** 10.3390/cancers17122029

**Published:** 2025-06-18

**Authors:** Amani A. Bashir, Mohamed M. El-Zaheri, Ahmad A. Bashir, Luma Fayyad, Aiman H. Obed, Dima Alkam, Abdalla Y. Bashir

**Affiliations:** 1Department of Pathology, University of Iowa Hospitals & Clinics, Iowa City, IA 52246, USA; amani-bashir@uiowa.edu; 2Department of Endocrinology, Ibn Sina University for Medical Sciences & Jordan Hospital, Amman 11152, Jordan; 3Department of Surgery, Ibn Sina University for Medical Sciences & Jordan Hospital, Amman 11152, Jordanaimanobed@hotmail.com (A.H.O.); 4Department of Pathology, Ibn Sina University for Medical Sciences & Jordan Hospital, Amman 11152, Jordan; l_fayyad@yahoo.com (L.F.); dmdmalk@outlook.com (D.A.)

**Keywords:** papillary thyroid microcarcinoma, incidental papillary thyroid microcarcinoma, non-incidental papillary thyroid microcarcinoma, total thyroidectomy, thyroid lobectomy, active surveillance

## Abstract

**Highlights:**

PTCs comprised 86.7% of all thyroid malignancies, and 36.2% were PTMCs.IPTMC occurred in 79.6% of patients and NIPTMC in 20.4%.NIPTMC was defined as carcinoma presenting as a thyroid nodule and/or abnormal lymph nodes occurring within normal thyroid tissue, without associated pathology.NIPTMCs were significantly associated with aggressive features, younger age, larger size, and multifocality.Management strategy includes the following:
a.Aggressive treatment is required for PTMCs associated with high-risk features.b.The extent of surgery in IPTMC is governed by the associated thyroid pathology.c.Physician-patient-shared decision-making in low-risk PTMCs can avoid overtreatment with TL or AS as options for management.

**Simple Summary:**

The incidence of papillary thyroid carcinoma has increased dramatically, mainly due to the increased detection of papillary thyroid microcarcinoma with the increased use of head and neck imagining and fine-needle aspiration cytology. Disease mortality rates, however, remain unchanged. This has led to the de-escalation of management recommendations for these lesions, including more limited and minimally invasive surgery and active surveillance as viable options. Although most papillary thyroid microcarcinomas show indolent behavior, a few cases behave more aggressively. We analyzed papillary thyroid microcarcinoma cases over a ten-year period and compared disease characteristics and outcomes of incidental vs. non-incidental disease. On an individual patient basis, management remains challenging given the absence of reliable clinical criteria for preoperative risk stratification. The majority of patients opt for immediate surgical management.

**Abstract:**

Background/Objectives: With evolving guidelines favoring de-escalation in the management of papillary thyroid microcarcinoma (PTMC), options such as active surveillance and minimally invasive procedures are now considered for patients with low-risk disease. However, a subset of PTMCs—particularly non-incidental cases—may exhibit aggressive behavior. This study compares disease characteristics and outcomes between incidental and non-incidental PTMCs over a 10-year period. Methods: This is a single-center retrospective comparative analysis utilizing a prospectively collected database of patients referred for thyroid surgery. Results: Papillary thyroid carcinoma accounted for 86.7% of thyroid malignancies, with PTMC comprising 36.2% (137 patients). Incidental PTMC represented 109 out of 1012 patients undergoing surgery for benign thyroid disease (10.8%). Non-incidental PTMC (NIPTMC), diagnosed preoperatively and presenting clinically without coexisting thyroid disease, was identified in 28 patients (20.4%). NIPTMCs were more frequently associated with high-risk features (75% vs. 10.1%, *p* = 0.004), including extrathyroidal extension (21.43% vs. 7.3% *p* = 0.0015), positive central lymph nodes (21.43% vs. 2.8%, *p* = 0.0291), positive lateral lymph nodes (28.6% vs. 0% *p* = 0.012), and lymphovascular invasion (3.6% vs. 0%). Multifocal PTMC was seen in 37 patients (27%), of which 27 had bilobar disease. Multifocal tumors had a higher likelihood of high-risk features (48.6% vs. 14%, *p* = 0.007). NIPTMC was a significant predictor of multifocality (*p* = 0.0098). All patients underwent surgery, none opted for active surveillance. **Conclusions:** NIPTMC is more often associated with high-risk features and multifocality, necessitating more extensive surgery. These findings emphasize the need for careful preoperative risk stratification to guide individualized management.

## 1. Introduction

Papillary thyroid carcinoma (PTC) stands as the most prevalent endocrine malignancy both nationally and globally, with a rising incidence over the last decade. This increased incidence is largely attributed to the increased detection of papillary thyroid microcarcinoma (PTMC) because of the more prevalent use of head and neck imaging and ultrasound-guided fine-needle aspiration cytology (US-FNAC), along with the increased awareness of such lesions [1,2,3,4]. Sosa et al. reported a 16% annual increase in US-FNAC usage in the United States, leading to a higher detection rate of PTC and a 31% rise in thyroid surgeries [5]. Nevertheless, the stable mortality rate from thyroid cancer raises concerns about overdiagnosis and overtreatment. Consequently, this has resulted in a shift to de-escalation in the management of these lesions, as reflected in the American Thyroid Association (ATA) guidelines recommending lobectomy and active surveillance (AS) as acceptable treatment options in selected patients [6,7], as well as, most recently, in the ATA’s consensus statement regarding the clinical use of minimally invasive ablative procedures, which stated that thermal ablation may be used safely in PTMC with no high-risk features, emphasizing that more data is needed for larger tumors [8].

Notably, AS of low-risk PTMC has been initiated in Japan based on the high incidence of both latent thyroid carcinoma in autopsy studies, reaching up to 36%, and small PTCs detected in mass screening studies compared to the prevalence of clinical carcinomas [9,10,11]. In a 10-year AS study of 1235 patients with low-risk PTMC, Miyauchi et al. [9] reported progression in the form of size enlargement by ≥3 mm and the appearance of nodal metastasis in only 8% and 3.8%, respectively. They noted that PTMCs were most unlikely to grow in older patients more than 60 years of age, in contrast to clinical PTC, suggesting that these patients are the best candidates for AS. Importantly, rescue surgery performed after signs of progression have been detected was not associated with a significant increase in recurrence rates. Furthermore, the total cost associated with immediate surgery, including the potential need for salvage surgery over 10-year period, was approximately four times higher than the cost of managing the disease with AS over the same timeframe. Their findings support AS as a safe and cost-effective first-line management strategy for patients with low-risk PTMC [9].

As per previous WHO classifications, the traditional and widely accepted definition of papillary thyroid microcarcinoma is a papillary carcinoma measuring 1 cm or less, and it has been reported as a distinct subtype [12]. However, due to the identification and reporting of biologically diverse types of PTMC, the most recent 2022 World Health Organization (WHO) classification no longer recognizes PTMC as a distinct subtype. Instead, it mandates the classification of these tumors based on their histomorphologic features, with an emphasis on the molecular profile [13]. The concept of low-risk neoplasm, along with a histologic and molecular-based grading system, helps guide personalized therapeutic decisions for patients at different levels of risk [13].

Although most PTMCs exhibit an indolent behavior, a subset of these tumors presents with high-risk aggressive features that are typically identified through final histopathologic examination post-surgery. The absence of reliable clinical criteria for preoperative risk stratification can result in insufficient or delayed treatment for these cases [4,14]. Moreover, current thyroid guidelines [6,7] do not endorse FNAC in sub-centimeter nodules unless certain clinical or ultrasonic features are present. Park et al. [15] concluded that PTMC should not be underestimated, as it can act as large PTC; they reported nodal metastasis in 34.9% of their patients. Yan et al. reported that, while small PTMCs are diagnosed incidentally, they can lead to regional lymph node involvement, which should be considered in the clinical management of these patients [16]. In addition, the frequent multifocality and bilaterality of PTMC dictate a tailored surgical approach involving patients in shared decision-making.

Clinically, PTMC can be incidental (IPTMC), referring to lesions discovered incidentally, often in thyroid resection specimens or non-incidental or primary (NIPTMC), referring to lesions diagnosed pre-surgically as palpable nodules or in conjunction with clinically apparent lymphadenopathy or distant metastasis. NIPTMCs more often present with aggressive features and can be associated with a worse prognosis [3,17,18,19].

This study aims to present the institution’s experience with incidental and non-incidental PTMC, including a comparative analysis of incidence, updated 2022 WHO histologic classification, clinicopathological characteristics, and therapeutic approaches, and assess the impact of evolving management guidelines on surgical practice alongside a review of the current literature.

## 2. Materials and Methods

This study retrospectively reviews the database of patients referred for thyroid surgery at Jordan Hospital/Ibn Sina University of Medical Sciences, a tertiary care academic medical center, from January 2013 to December 2022. This study includes consecutive patients with papillary thyroid carcinoma over the age of 18 and excludes patients undergoing combined thyroid and parathyroid surgery and patients with familial MENII syndrome. The procedures in the current study were followed and assessed by the institutional review committee in accordance with the Declaration of Helsinki and its later amendments. Informed consent was obtained from all patients before the initiation of this study.

Initial data collection encompassed demographic information such as age, sex, and nationality. Clinical evaluations included growth rate, compression symptoms, hoarseness of voice, presence of hypo- or hyperthyroidism, history of radiation exposure, family history of thyroid disease or malignancy, and neck physical examination findings, specifically the presence of palpable thyroid nodules or lymph node enlargement.

Laboratory investigations included measurements of TSH, T3, T4, thyroglobulin, and thyroid autoantibodies. Preoperative ultrasonography was performed to characterize thyroid nodules, assess lymph node status, and US-FNAC of solitary, dominant, or any suspicious nodules and lymph nodes. FNAC results were categorized according to the Bethesda system [20,21]. Scintigraphy was conducted in selected cases with suspected thyrotoxic nodules. Informed consent and patient counseling covered treatment plans, intraoperative decisions, and patient preferences. All patients underwent surgical resection, with therapeutic central neck dissection for clinically or ultrasound-positive or suspicious nodes and modified neck dissection for positive lateral nodes. None of the patients opted for active surveillance (AS).

Histopathological examination was performed by two board-certified pathologists experienced in thyroid pathology from Jordan Hospital (Amman, Jordan). Thyroid specimens resected for benign thyroid disease were thoroughly examined for incidental microcarcinomas (≤1 cm). Over a one-month period, the first author, a surgical pathologist and cytopathologist from the University of Iowa (Iowa City, IA, USA) performed a blinded re-evaluation of all original slides of PTMC surgical cases diagnosed by local pathologists during the period of the study (2013–2022) and classified all PTMCs based on the updated 2022 WHO classification of thyroid neoplasms [13]. Pathologic examination of tumors included assessments of histologic subtype, size, site, margins, bilaterality, multifocality, the presence or absence of extrathyroidal extension (ETE), and lymphovascular invasion.

The median patient follow-up duration was five years. The recurrence surveillance protocol included clinical examination, thyroid ultrasound, and laboratory testing for thyroglobulin, antithyroglobulin antibodies, and TSH every three months for the first year, then every sixth months for two additional years and once a year thereafter.

Based on our patient cohort, PTMCs were classified into two groups: 1. Incidental PTMC (IPTMC) represents tumors identified incidentally in patients referred for thyroid surgery for benign thyroid disease, including multinodular goiter, Hashimoto thyroiditis, Graves’ disease, or adenoma. These tumors were predominantly identified postoperatively through the pathological examination of surgical specimens. 2. Non-incidental or primary PTMC (NIPTMC) includes tumors that present clinically as thyroid nodules or abnormal lymph nodes, are diagnosed preoperatively through clinical evaluation, imaging, or fine-needle aspiration cytology (FNAC), and arise in otherwise normal thyroid tissue without associated thyroid pathology (Figure 1).

A comparative analysis of incidental versus NIPTMCs was performed, focusing on incidence, demographic data, FNAC results, incidence of associated benign thyroid disease, presence of high-risk aggressive features including local ETE, positive central or lateral cervical lymph nodes, lymphovascular invasion, and aggressive histologic subtypes.

Additionally, multifocality was studied, and a comparative analysis was conducted to assess unifocal versus multifocal disease. The analysis also included an assessment of predictors of multifocality and the incidence of bilaterality. The number of patients with PTMCs and the type of surgery were compared between the study’s first and second five-year periods.

Statistical analysis was conducted using IBM SPSS Statistics for Windows, version 25 (released in 2017, IBM Corp., Armonk, NY, USA). Means are expressed with standard deviation, and a *p*-value of <0.05 is considered statistically significant. Appropriate statistical tests, such as the *t*-test, Chi-Square test, and ANOVA, were utilized to derive *p*-values and analyze the data.

## 3. Results

Over the ten-year duration of the study, there were 378 cases (86.7%) of PTC out of 436 patients diagnosed with thyroid malignancy. PTMC T1a accounted for 137 cases (36.2%), while stages T1b, T2, and T3 PTC comprised 77 (20.4%), 110 (29.1%), and 54 (14.4%) cases, respectively. Within the PTMC (T1a) category, 28 cases were classified as NIPTMC and 109 cases as IPTMC (Table 1).

A total of 1012 patients underwent surgical resection for benign thyroid disease, of which 109 patients (10.8%) were found to have IPTMC. IPTMC was identified in 67 of 770 patients with multinodular goiter (MNG) (8.7%), 21 of 101 patients with Hashimoto thyroiditis (HT) (20.8%), 11 of 62 patients with Graves’ disease (17.74%), and 10 of 79 patients with adenomas (12.7%) (Table 1). In these patients, surgery was performed for non-cancer reasons. In MNG, surgery was performed due to compression symptoms, complications, or retrosternal extension. In Graves’ disease, it was performed due to ophthalmopathy, failure or complication of medical treatment, or the presence of a nodular gland. Nodular HT occurred in nine patients (28.6%).

The histologic subtypes of PTMC cases (based on the updated 2022 WHO classification of thyroid neoplasms) are detailed in Table 2. The classic subtype was significantly the most prevalent in NIPTMC cases (89.3%). Similarly, the classic subtype was the most common in IPTMC cases (52.3%), followed by the minimally invasive non-encapsulated/circumscribed follicular variant of papillary carcinoma (35.8%), which was significantly more prevalent in IPTMC than in NIPTMC (*p* = 0.001). Aggressive subtypes (tall cell, columnar cell, hobnail, diffuse sclerosing, and solid) were not observed.

A comparative analysis of demographic and clinicopathological features of IPTMC and NIPTMC is presented in Table 3. IPTMC was significantly more prevalent (79.6%) compared to NIPTMC (20.4%) (*p* < 0.045). There was no significant gender difference between the two types (*p* = 0.324). The mean age was significantly younger in NIPTMC cases (37.14 ± 14.28 years) compared to IPTMC cases (44.15 ± 13.43 years) (*p* = 0.0001). Additionally, NIPTMC was significantly more common in patients younger than 55 years (*p* < 0.05). IPTMC was significantly more associated with MNG (*p* = 0.0005) and HT (*p* = 0.0054) than with Graves’ disease (*p* = 0.1171) and adenomas (*p* = 0.5513) (Table 3).

High-risk aggressive features were observed in 32 out of 137 patients (23.36%) and were significantly more common in NIPTMC than in IPTMC (*p* = 0.004), including local extrathyroidal extension (ETE) (21.43% vs. 7.3%), positive central nodes (21.43% vs. 2.8%), and positive lateral cervical lymph nodes (28.6% vs. 0%). The diagnosis of malignancy was made preoperatively clinically, by FNAC, or the presence of nodal metastasis (Table 3).

Most patients (100 patients) had unifocal disease (73%); 85 (85%) were IPTMCs, and 15 (15%) were NIPTMCs. Multifocal disease (MF) was present in 37 patients (27%), with 24 IPTMC cases (64.8%) and 13 NIPTMC cases (35.2%). Bilateral PTMCs occurred in 27 patients with MF-PTMCs (73%), as shown in (Table 4).

The average maximum tumor diameter was 0.442 cm in unifocal tumors and 0.78 cm in multifocal tumors (*p* = 0.0054). Multifocality was more common in association with tumors larger than 5 mm; larger tumors (>5 mm) were present in 43 (43%) unifocal cases and 29 (78.4%) multifocal cases (*p* < 0.0002). There were no significant gender or age differences between unifocal and MF-PTMCs (Table 4).

Predictors of multifocality included NIPTMC (significantly, *p* = 0.0098), as well as MNG and HT (non-significantly), compared to Graves’ disease and adenomas. Aggressive features were significantly more common in MF-PTMCs, with an incidence of 48.6% compared to 14% in unifocal disease (*p* < 0.007).

Total thyroidectomy was performed in all cases of NIPTMCs and MF-PTMCs (100%), compared to 88% in IPTMCs and unifocal PTMCs (*p* < 0.001) (Table 4). This included eight completion thyroidectomies requested by female patients aged 32–36 years due to concerns about malignancy risk.

Twelve lobectomies were performed in cases of IPTMC (Table 3). These included eight patients with multinodular goiter, three with Hashimoto thyroiditis, and one with a follicular adenoma. Upon histopathologic evaluation, all tumors were classified as low-risk. Accordingly, following patient counseling and a shared decision-making process, the patients endorsed the conservative management approach.

During the first five years of the study (2013–2017), 67 PTMC cases (48.9%) were identified, compared to 70 cases (51.1%) in the second five years (2018–2022) (*p* = 0.071). Total thyroidectomy (TT) and thyroid lobectomy (TL) were performed in 57 (85.1%) and 10 (14.9%) patients, respectively, during the first period, compared to 68 (97.1%) and 2 (2.9%) patients during the second period (*p* = 0.0022). Upon follow-up, all patients with PTMC were alive, with no disease recurrence. Recurrence occurred in twelve patients with higher-stage PTC (T1b and above), including ten female and two male patients. One patient succumbed to dedifferentiated recurrent PTC.

Table 5 presents the comparative demographics and clinicopathological features of PTMC (T1a) and T1b PTC. There were no significant differences in gender, age, multifocality, or surgical treatment. T1b PTCs were more frequently associated with high-risk aggressive features (56.65%) compared to PTMC (T1a) (23.35%), *p* = 0.001), indicating advancing disease.

## 4. Discussion

PTC is the most common thyroid malignancy. Kaliszewski et al. [22] reported a fourfold increase in PTMC incidence. In this study, PTC accounted for 86.7% of thyroid malignancies, with PTMC comprising 36.2% of cases. The literature describes two clinically distinct types of PTMC, IPTMC and NIPTMC, which differ in clinical presentation, characteristics, and behavior [19,22].

Lombardi et al. and Durante et al. reported that IPTMC constituted the majority of PTMC cases, at 75.5% and 77%, respectively [23,24]. Elliot et al. [17] found an equal distribution of IPTMC and NIPTMC, whereas Kaliszewski et al. [22] reported that NIPTMC comprised 66.67% of PTMCs. In our study, 79.6% of PTMCs were IPTMCs identified incidentally in patients referred for surgery for benign thyroid disease and diagnosed primarily in post-surgical specimens (Table 1), while NIPTMCs represented 20.4%. These tumors were diagnosed preoperatively and presented clinically as thyroid nodules or abnormal lymph nodes without associated thyroid pathology (Figure 1). The variation in the reported incidence of PTMC is likely related to different geographic populations, with some cohorts arising from endemic goiter areas [17].

The incidence of IPTMC in patients operated on for benign thyroid disease was 10.8%, compared to 9.3%, 10.41%, 16.5%, and 15.6% reported by Miccoli et al., de Carlos et al., Slijepcevic et al., and Smith et al. [25,26,27,28], respectively. IPTMC was significantly associated with MNG (*p* = 0.0005) and HT (*p* = 0.005) compared to Graves’ disease (*p* = 0.1171) and follicular adenoma (*p* = 0.5513) (Table 3). Similar findings were reported by other authors [27,28]. Smith et al. [28] found a significant association of IPTMC with MNG and toxic nodular goiter compared to Graves’ disease (*p* = 0.01), while Slijepcevic et al. [27] reported a higher incidence in HT patients and a lower incidence in toxic nodular goiter and Graves’ disease.

The association between HT and PTC has been extensively investigated. Bircan et al. [29] reported a 39% overall incidence of HT in PTMC cases, with concomitant occurrence ranging from 0.5% to 58% [29]. Graceffa et al. [30] reported a 40.2% association with the nodular variant of HT and 8.1% with the diffuse variant (*p* < 0.001), hypothesizing that the link may be due to an autoimmune response that develops alongside an antitumoral immune response. Nodular HT occurred in (42.9%) in the present study. Anderson et al. [31] reported nodular HT and PTMC in 42.7% and 16%, respectively. Miccoli et al. [25] and Paparodes et al. [32] found that incidental thyroid carcinoma was significantly more frequent in euthyroid patients than in thyrotoxic or fully hypothyroid HT patients (*p* < 0.03).

Our study shows no significant gender difference between incidental and non-incidental PTMC (*p* = 0.324). NIPTMC patients were significantly younger, with a mean age of 37.14 ± 14.28 years compared to 44.15 ± 13.43 years in IPTMC cases (*p* = 0.0001), consistent with Elliott et al. [17]. Miccoli et al. and de Carlos et al. [25,26] reported no significant age or sex differences, while Smith et al. [28] and Lee et al. [16] found significant associations of young age and male sex with IPTMC.

PTMC generally has an indolent course with an excellent prognosis. However, an increasing number of studies have identified an aggressive subset of PTMC with high-risk pathologic features that negatively impact clinical outcomes and overall survival. In our study, high-risk features were present in 23.36% of cases, being significantly more common in NIPTMC than IPTMC (75% vs. 10.1%, *p* = 0.004) (Table 3). Retrospective analyses by Lee et al. [14] and Al-Qurayshi et al. [33], using the National Cancer Database, found that 20% and 19% of PTMCs, respectively, presented with high-risk aggressive features, including ETE, central or lateral node involvement, lymphovascular invasion, and distant metastasis.

Aggressive histologic subtypes of PTMC have been associated with increased ETE and nodal involvement, as reported by Lee et al., indicating that these variants should be considered to be important factors when determining the extent of surgical intervention. In a study of 148 PTMC cases, Ghossein et al. found that larger nodal metastases (>1 cm) and aggressive histologic subtypes were associated with worse recurrence-free survival [34]. Our cohort did not include aggressive histologic subtypes such as tall cells, columnar cells, hobnail, diffuse sclerosing, and solid subtypes. The classic subtype was the most common in both NIPTMC (89.3%) and IPTMC (52.8%) cases, consistent with previous reports [25,35]. The minimally invasive non-encapsulated (circumscribed) follicular variant of papillary thyroid carcinoma (FVPTC) was significantly more common in IPTMC (35.8%) (Table 2).

PTMC shares similar molecular and immunophenotypic characteristics with larger PTCs. Park et al. reported comparable frequencies of the BRAFV600E mutation in PTMC and larger PTCs (65.6% vs. 67.2%) [15]. Recent studies suggest that telomerase reverse transcriptase (TERT) promoter mutations, which correlate with aggressive behavior and poor outcomes, are more common in aggressive histologic subtypes. However, their occurrence in PTMC is extremely low. Yang et al. reported a TERT promoter mutation frequency of 0.5% in PTMC compared to 5.8% in PTC > 1 cm [36]. Currently, molecular testing is primarily used for larger PTC and its applicability to PTMC remains unclear. Further studies are needed to explore its prognostic role in PTMC [37,38,39]. However, if thyroidectomy is needed or preferred, molecular testing is unnecessary [40]. None of our cases underwent molecular testing, as these tests are not readily available for routine clinical use due to the prohibitive cost and lack of insurance coverage.

In our series, multifocal disease (MF) was present in 29.6% of PTMC patients who underwent total thyroidectomy (37 out of 125); in our study, 73% were bilateral, with an average of 2.9 tumor foci. This aligns with published reports indicating multifocality ranging from 18 to 87% and bilaterality from 13 to 56% [14,19,26,41,42]. So et al. [41] noted that the likelihood of contralateral lobe involvement increases with the increase in number of foci in one lobe, at 30.2% with two foci and 46.2% with three or more foci.

Similarly to previously published reports [42], our analysis concluded that multifocal PTMCs were more frequently associated with high-risk aggressive features compared to unifocal tumors (48.6% vs. 14%, *p* = 0.007) and had a larger average maximum tumor diameter (*p* = 0.0054) (Table 4). Varshney et al. found that only ETE had a significant association with lymph node metastases (*p* > 0.05) [43]. In contrast, Dirikoc et al. identified MF and ETE as significant factors for lymph node metastasis [44].

Age and sex were not significant predictors of multifocality in our study, consistent with other reports [41]. Elbasan et al. [42] also reported no differences in age, sex, and BRAF mutation positivity between unifocal and bilateral tumors. Park et al. [45] identified age as a determinant of overall survival (OS) and disease-specific survival (DSS), while sex and multifocality were also prognosticators of OS. Our analysis showed that NIPTMC had a significantly higher association with multifocality, which was more common in MNG and HT (non-significantly) compared to Grave’s disease and adenomas (Table 4).

The number of PTMC cases remained stable over the two five-year periods of our study (2013–2017 vs. 2018–2022, *p* = 0.071), contrary to the literature reporting an increased incidence over time [22,26]. This stability may reflect the impact of new ATA guidelines limiting FNAC in sub-centimeter nodules and possibly the effect of the 2020 pandemic on patient numbers.

The optimal management of PTMC continues to be a subject of significant debate due to variability in tumor presentation and behavior, challenging preoperative diagnosis, and variable incidence of multifocality, bilaterality, and aggressive features. Aggressive PTMCs often lack clinical signs and reliable diagnostic criteria for preoperative identification. Distinguishing between aggressive and indolent PTMC remains difficult [3,4].

Current ATA guidelines recommend TL or active surveillance for selected PTMC patients [6]. Despite this, TT remains the most common surgical approach, performed in 82.63%, 83%, and 73.9% of PTMCs reported by Al-Qurayshi et al. [33], Lee et al. [14], and Park et al. [45], respectively. The incidence of TT increased by 12% per year in the U.S., while lobectomies increased by 1% per year [5]. Advocates of TT cite frequent multifocality, bilaterality, and easier follow-up, while proponents of TL highlight lower surgical complication rates, especially for low-volume surgeons [46], and avoidance of lifetime thyroxine replacement therapy [47] and hypoparathyroidism risk [48]. In our study, fewer TLs (2.9%) and more TTs (97.1%) were performed in the second period compared to the first (14.9% TLs and 85.1% TTs) (*p* = 0.0007), indicating no impact from the 2017 ATA guidelines on our surgical practice [6].

The management strategy at Jordan Hospital Medical Center includes the following:High-risk PTMCs: PTMCs with high-risk features such as ETE, lymph node metastasis, aggressive histologic subtype, and proximity to the trachea or recurrent laryngeal nerve require aggressive surgical management [49], including total thyroidectomy, lymph node dissection for clinical or US-positive nodes, and radioactive iodine ablation.Benign thyroid disease: Patients with presumed benign thyroid disease are referred for surgery based on specific indications related to the thyroid pathology. Total thyroidectomy is often required for associated conditions like MNG, toxic nodular goiter, Graves’ disease, and Hashimoto thyroiditis (HT). PTMC diagnosis in these cases is typically made postoperatively through the pathological examination of thyroidectomy specimens.Low-risk PTMCs: For preoperatively diagnosed low-risk PTMCs, including unifocal intrathyroidal tumors with clinically negative nodes, management is determined through a physician–patient shared decision-making process, considering clinical evidence, patient preferences, beliefs, comorbidities, and available resources [45,50,51]. AS or TL are viable options.

Miyauchi et al. [9] reported that, based on the accumulation of long-term data, AS for low-risk PTMC is a safe management strategy and offers greater benefits than immediate surgery for both patients and society. Unfortunately, none of our patients opted for AS. In our experience, cultural perceptions and heightened anxiety surrounding cancer contribute to a preference for definitive surgical management, potentially limiting the adoption of more conservative approaches. Additionally, the success of active surveillance relies heavily on patient adherence to follow-up and recurrence-monitoring protocols. However, concerns regarding long-term compliance may deter some patients from opting for this management strategy. Ultimately, increasing awareness and education among endocrinologists, surgeons, patients, and the public is essential to promote AS as a safe and effective management option for low-risk disease.

As previously reported by Bashir et al. [50], our experience shows that patients favor TT due to fear of cancer recurrence or potential future surgeries and for ease of follow-up, especially for international patients seeking treatment with minimal delays. Preoperatively, the fear of malignancy was the reason for choosing surgery in 35.1% of patients, avoiding FNAC in 20.2% of patients, and refusal of AS. Patient preference-based decisions for TT vs. TL were less guideline-concordant than decisions based on frozen section (21% vs. 79%), with more TTs performed (74.4% vs. 41%, *p* = 0.001). Notably, in a global healthcare setting, molecular testing is not readily available due to cost and the lack of insurance coverage. Improving patient understanding of the disease and up-to-date management options is crucial for more effective shared decision-making [51]. The ATA risk stratification for PTMC aids clinical decision-making [4].

This study has several limitations. It is a retrospective analysis from a single institution. The relatively small sample size of the NIPTMC group (*n* = 28) represents a limitation that may impact the statistical power and reduce the generalizability of our findings. None of the patients underwent AS; therefore, disease progression data is unavailable. Furthermore, the lack of an AS cohort limits our ability to compare outcomes with more conservative treatment strategies. Molecular testing was not performed on any of our cases. Although molecular testing can assist in identifying high-risk mutations, it is not currently accessible for routine clinical use in Jordan. The high cost of molecular testing continues to pose a significant financial barrier to widespread use, particularly in resource-constrained settings.

## 5. Conclusions

PTMC is the most common thyroid malignancy. In our study, PTC comprised 86.7% of all thyroid malignancies, with PTMC occurring in 36.2% of cases; 79.6% (*n* = 109) were IPTMC and 20.4% (*n* = 28) were NIPTMC. While generally indolent, some cases are associated with aggressive behavior and high-risk features, requiring more aggressive surgical intervention, especially in NIPTMC as demonstrated in our study and by others [3,17,18,19]. The challenge lies in identifying and applying prognostic features in preoperative settings. The outcomes for patients in the current study were favorable, with no mortality or recurrence during the follow-up period. However, most patients underwent total thyroidectomy. Multi-center clinical trials are needed to help with presurgical disease progression risk stratification in PTMC and to determine the best treatment strategies, prognosis, and outcomes. Shared decision-making between physicians and patients is crucial for optimal outcomes.

## Figures and Tables

**Figure 1 cancers-17-02029-f001:**
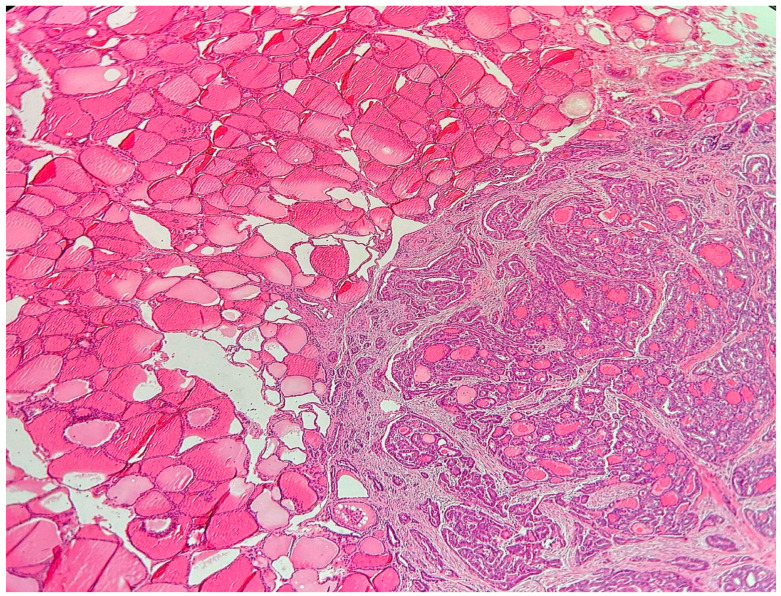
Non-incidental papillary thyroid microcarcinoma (NIPTMC) arising in a background of normal thyroid tissue, 20× H&E.

**Table 1 cancers-17-02029-t001:** Patient cohort and pathological diagnoses.

(1) Begin Thyroid Disease (1012)		IPTMC ** Incidence
1. Nodular Goiter	770 (76.1%)	67 (8.7%)
2. Hashimoto Thyroiditis (HT)	101 (10%)	21 (20.8%)
3. Graves’	62 (6.1%)	11 (17.74%)
4. Adenoma	79 (7.8%)	10 (12.7%)
Total	1012 (100%)	109 (10.8%) ***
**(2) Thyroid Malignancies (436)**		
**(1) Papillary Carcinoma (PTC) ***	**Number 378 (86.7%)**	
T1: T1a: < 1 cm (PTMC)	137 (36.2%)	
Incidental	109	
Non-Incidental	28	
T1b: >1–≤2 cm	77 (20.4%)	
T 2: >2 cm–≤4 cm	110 (29.1%)	
T 3: > 4 cm	54 (14.3%)	
Total	378 (100%)	
**(2) Other Malignancies**	**Number 58 (13.3%)**	
Follicular Carcinoma	23	
Oncocytic (Hurthle Cell) Carcinoma	8	
Medullary Carcinoma	12	
Lymphoma	4	
Anaplastic Carcinoma	2	
Well Differentiated Tumor of	8	
Undetermined Malignant Potential		
Renal Metastasis	1	
Total	58	
All Malignancies	436 (100%)	

* PTC: papillary thyroid carcinoma; ** IPTMC: incidental papillary thyroid microcarcinoma, *** Included in T1a PTC.

**Table 2 cancers-17-02029-t002:** Histologic subtypes of papillary thyroid microcarcinoma (PTMC).

Clinical Presentation.	Histologic Subtype of PTMC					
1. IPTMC * Associated with	Classic Subtype	Infiltrative Follicular Subtype	FVPTC *** Encapsulated (Minimally Invasive)	FVPTC Non-Encapsulated/Circumscribed (Minimally Invasive)	Oncocytic Subtype	Total Number
**Multinodular Goiter**	38 (56.7%)	0 (0%)	5 (7.5%)	23 (34.3%)	1 (1.5%)	67 (100%)
**Hashimoto Thyroiditis**	8 (38%)	4 (19%)	0 (0%)	9 (43%)	0 (0%)	21 (100%)
**Graves’ Disease**	6 (54.5%)	0 (0%)	0 (0%)	4 (36.4%)	1 (9.1%)	11 (100%)
**Adenoma**	5 (50%)	0 (0%)	1 (10%)	3 (30%)	1 (10%)	10(100%)
1. Total: IPTMC	57 (52.3%)	4 (3.7%)	6 (5.4%)	39 (35.8%)	3 (2.8%)	109 (100%)
**2. NIPTMC ****	25 (89.3%)	1 (3.6%)	0 (0%)	2 (7.1%)	0 (0%)	28 (100%)
***p* Value**	0.002	0.4801	0.1515	0.001	0.3751	0.0001
	<0.05	>0.05	>0.05	<0.05	>0.05	<0.5

* IPMTC: incidental papillary thyroid microcarcinoma; ** NIPTMC: non-incidental papillary thyroid microcarcinoma; *** FVPTC: follicular variant of papillary thyroid carcinoma (currently distinct from papillary thyroid carcinomas due the prevalence of RAS-like mutations). Classification is based on the 2022 WHO Classification of Thyroid Neoplasms.

**Table 3 cancers-17-02029-t003:** Demographic and clinicopathological features of patients with papillary thyroid microcarcinoma (PTMC).

Type		Incidental	Non-Incidental (Primary)	*p* Value	Status p. v
**Number of Cases**		109 (79.6%)	28 (20.4%)	0.045	<0.05
**Gender**	F	85 (77.9%)	20 (71.43%)	0.324	>0.05
	M	24 (22.1%)	8 (28.57%)		
**Age (Y)**	(1) Average Y	44.153 ± 11.28 y	37.14 ± 13.43y	0.0001	<0.05
	(2) <45 Y	50 (45.9%)	20 (71.43%)	6.93	<0.05
	>45 Y	59 (54.1%)	8 (28.57%)		
	<55 Y	86 (78.9%)	27 (96.43%)		
	>55 Y	23 (21.1%)	1 (3.57%)		
**Associated Pathology**					
	Multinodular Goiter	67 (61.5%)	*	0.0005	<0.05
	Hashimoto Thyroiditis	21 (19.3%)	*	0.0054	<0.05
	Adenoma	10 (9.2%)	*	0.5513	>0.05
	Graves’ disease	11 (10%)	*	0.1174	>0.05
**Nationality**					
Local		60 (55%)	16 (57.1%)	0.35	>0.05
International		49 (45%)	12 (42.9%)		
**FNAC BVI/BV ***					
	Yes	48 (44%)	24 (85.71%)	0.001	<0.05
	No	61 (56%)	4 (14.29%)		
**Aggressive Features**					
Extrathyroidal Extension	Yes	8 (7.3%)	6 (21.43%)	0.0015	<0.05
No	101 (92.7%)	22 (78.57%)			
Positive Central nodes	Yes	3 (2.8%)	6 (21.43%)	0.0291	<0.05
	No	106 (97.2%)	22 (78.59%)		
Positive Lateral Nodes	Yes	0 (0%)	8 (28.6%)	0.012	<0.05
	No	109 (100%)	20 (71.4%)		
Lymphovascular invasion	Yes	0 (0%)	1 (3.6%)	_	>0.05
	No	109 (100%)	27 (96.4%)	_	
Aggressive FeaturesTotal	Yes	11 (10.1%)	21 (75%)	0.004	<0.05
	No	98 (89.9%)	7 (25%)		
**Total Thyroidectomy**		97 (88.1%)	28 (100%)	0.0303	<0.05
**Total Lobectomy**		12 (11.9%)	0 (0%)		

* FNAC BVI or/BV: fine-needle aspiration cytology Bethesda VI or Bethesda V.

**Table 4 cancers-17-02029-t004:** Papillary thyroid microcarcinoma (PTMC) multifocality (MF) and clinicopathological features.

Type	Unifocal	Multifocal (MF)	*p*-Value
(1) Number	100 (73%)	37 (27%)	<0.002
(2) Average maximal tumoral diameter	0.442 cm	0.78 cm	0.0054
(3) Size ≥ 5 mm	43 (43%)	29 (78.4%)	<0.0002
(4) Site			>0.1367
Unilobar (Right or Left)	97 (97%)	Bilobar 27 (73%)	
Isthmus	3 (3%)	0%	
(5) Gender			
F	76 (76%)	27 (73%)	>0.7248
M	24 (24%)	10 (27%)	
(6) Age:			
F	43.46 ± 11.49	45.55 ± 13.45	>0.6323
M	42.95 ± 12.18	43.3 ± 17.13	
(7) FNA BVI or/BV	42 (42%)	21 (56.8%)	>0.1294
(8) Predictors of MF:			
IPTMC *	85 (85%)	24 (64.8%)	0.0915
NIPTMC **	15 (15%)	13 (35.2%)	<0.0098
Nodular goiter	52 (52%)	15 (40.5%)	>0.2319
Hashimoto thyroiditis	14 (14%)	7 (18.9%)	>0.4642
Graves’ disease	10 (10%)	1 (2.7%)	>0.1640
Adenoma	9 (9%)	1 (2.7%)	>0.2048
(8) Aggressive features			
ETE	***		
Yes	8 (8%)	6 (16.21%)	>0.2371
NO	92 (92%)	31 (83.79%)	
Positive Central Nodes			
Yes	3 (3%)	5 (13.5%)	>0.0768
No	97 (97%)	32 (86.5%)	
Positive Lateral Nodes			
Yes	3 (3%)	6 (16.21%)	>0.0591
No	97 (97%)	31 (83.79%)	
Lymphovascular invasion			
Yes	0 (0%)	1 (2.7%)	>0.5307
No	0 (0%)	36 (97.3%)	
Aggressive features total			
Yes	14 (14%)	18 (48.6%)	<0.007
No	86 (86%)	19 (51.4%)	
(9) Total Thyroidectomy			
Yes	88 (88%)	37 (100%)	<0.001
No	12 (12%)	0 (0%)	

* IPTMC: incidental papillary thyroid microcarcinoma; ** NIPTMC: non-incidental papillary thyroid microcarcinoma; *** ETE: extrathyroidal extension.

**Table 5 cancers-17-02029-t005:** Comparative analysis between papillary thyroid microcarcinoma (PTMC) T1a and T1b papillary thyroid carcinoma.

Stage (Number of Cases)	Tla (137)		TIb (77)		*p* Value	Status p.v.
Gender F	103 (76.6%)		57 (74%)		0.607	>0.05
M	34 (23.4%)		20 (26%)			
Age <45 Y	73 (53.28%)		57 (74.1%)		0.0508	>0.05
≥45 Y	64 (46.72%)		20 (25.9%)			
≥55 Y	25 (18.24%)		11 (14.3%)			
<55 Y	112 (81.76%)		66 (85.7%)			
Aggressive Features						
(1) Extra thyroidal Extension					0.00053	<0.05
Yes	14 (10.2%)		21 (27.3%)			
No	123 (90.8%)		56 (72.7%)			
(2) Positive Central Nodes Yes	8 (5.8%)		11 (14.3%)		0.0281	<0.05
No	129 (94.2%)		66 (85.7%)			
(3) Positive Lateral Nodes Yes	9 (6.6%)		9 (11.69%)		0.8524	>0.05
No	128 (93.4%)		68 (88.31%)			
(4) Lymphovascular invasion Yes	1 (0.73%)		1 (1.37%)		0.5186	>0.05
No	136 (99.27%)		76 (97.4%)			
Total Yes	34 (23.35%)		44 (56.4%)		0.0001	<0.05
No	105 (76.65%)		34 (43.6%)			
Unifocal					0.6550	>0.05
Right	55 (40.14%)		28 (36.4%)			
Left	43 (30.66%)		20 (26%)			
Isthmus	3 (2.2%)		2 (2.6%)			
Total	100 (73%)		50 (64.9%)			
	one lobe	two lobes	one lobe	two lobes		
Multifocal	37		27		9.4880	>0.05
2 foci	6 (16.2%)	12 (44.4%)	5 (18.3%)	7 (25.9%)		
≥3 foci	4 (10.8%)	15 (55.6%)	6 (22.2%)	9 (33.4%)		
Total	10 (27.1)	27 (73%)	11 (40.7%)	16 (59.3%)		
Total Thyroidectomy	125 (91.24%)		77 (100%)		0.992	>0.05
Total Lobectomy	12 (8.76%)		0 (0%)			

## Data Availability

The data that support the findings of this study are available from the corresponding author upon reasonable request.

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
