# Peer review of "Papillary Thyroid Microcarcinoma in Thyroid Surgical Practice: Incidental vs. Non-Incidental: A Ten-Year Comparative Study"

_cancers, 2025, doi:10.3390/cancers17122029_

Round 1
Reviewer 1 Report
Comments and Suggestions for Authors
The authors in this study present the institution’s experience with incidental and non-incidental papillary thyroid microcarcinoma, including a comparative analysis of incidence, updated 2022 WHO histologic classification, clinicopathological characteristics, and therapeutic approach, and assess the impact of evolving management guidelines on surgical practice alongside a review of current literature.
The paper is scientifically satisfactory, rigorous and well documented. The title reflect the main subject of the manuscript, the abstract reflect the work described in the paper, the key words reflect the focus of the manuscript. The manuscript adequately describe the background, present status and significance of the study, and describe methods in adequate detail.
The study contributes significantly for surgical management of incidental and non-incidental papillary thyroid microcarcinoma.
The manuscript interpret the findings adequately and appropriately, highlighting the key points concisely, clearly and logically. The tables and figures are adequate. The bibliography is good.
Author Response
Comment 1:
The authors in this study present the institution’s experience with incidental and non-incidental papillary thyroid microcarcinoma, including a comparative analysis of incidence, updated 2022 WHO histologic classification, clinicopathological characteristics, and therapeutic approach, and assess the impact of evolving management guidelines on surgical practice alongside a review of current literature.
The paper is scientifically satisfactory, rigorous and well documented. The title reflects the main subject of the manuscript, the abstract reflect the work described in the paper, the key words reflect the focus of the manuscript. The manuscript adequately describes the background, present status and significance of the study, and describe methods in adequate detail.
The study contributes significantly for surgical management of incidental and non-incidental papillary thyroid microcarcinoma.
The manuscript interprets the findings adequately and appropriately, highlighting the key points concisely, clearly and logically. The tables and figures are adequate. The bibliography is good.
Response
Thank you very much for your positive feedback and for taking the time to review our manuscript. We are pleased to hear that you found the paper satisfactory and that no revisions were necessary. Your encouraging comments are greatly appreciated.
Reviewer 2 Report
Comments and Suggestions for Authors
Bashir et al. provide a manuscript about thyroid surgical practice in papillary thyroid microcarcinoma. The manuscript is written from thyroid surgeon's view. It is well known that the ATA guideline introducing active surveillance (AS) in PTMC raised many controversies and met with intensive resistance from many clinicians, especially thyroid surgeons. The provided manuscript reflects this resistance. I miss in the manuscript the comprehensive explanation, why active surveillance is safe. I do not consider the data given in the manuscript as an evident argument against active surveillance in PTMC. It should be clear for all thyroidologists that the preoperative presence of high risk features should exclude the option of AS. I am worried that " "the new ATA guideline had no impact on institutional surgical practice during the study period". My suggestion to the Authors would be to wonder why it happened and how should they act to change this bad practice. I am convinced that only coordinated efforts of the whole thyroidological community will help introducing active surveillance as an optimal and safe approach in PTMC.
Author Response
Comment 1:
Bashir et al. provide a manuscript about thyroid surgical practice in papillary thyroid microcarcinoma. The manuscript is written from thyroid surgeon's view. It is well known that the ATA guideline introducing active surveillance (AS) in PTMC raised many controversies and met with intensive resistance from many clinicians, especially thyroid surgeons. The provided manuscript reflects this resistance. I miss in the manuscript the comprehensive explanation, why active surveillance is safe. I do not consider the data given in the manuscript as an evident argument against active surveillance in PTMC. It should be clear for all thyroidologists that the preoperative presence of high risk features should exclude the option of AS. I am worried that " "the new ATA guideline had no impact on institutional surgical practice during the study period". My suggestion to the Authors would be to wonder why it happened and how should they act to change this bad practice. I am convinced that only coordinated efforts of the whole thyroidological community will help introducing active surveillance as an optimal and safe approach in PTMC.
Response:
We thank the reviewer for their thoughtful comments and the opportunity to clarify the scope and intent of our manuscript. Our study presents our institution’s experience with papillary thyroid microcarcinoma (PTMC), with a particular focus on the clinical differences between incidental and non-incidental PTMC (NIPTMC). As discussed in the manuscript, these subtypes have been reported to behave differently, with NIPTMC more frequently associated with high-risk features
We recognize that none of the patients in our cohort opted for AS. This may reflect a heightened cultural fear of cancer and a preference for definitive surgical management. We agree that only through coordinated efforts of the thyroidology community, along with education and increased awareness among physicians and the public, can we begin to shift the longstanding mindset favoring aggressive surgical interventions toward embracing more conservative management. This includes recognizing active surveillance (AS) as a safe approach for appropriately selected patients.
Importantly, the intent of our study is not to argue against AS, but rather to highlight that a subset of PTMC cases may exhibit more aggressive behavior. Given the limitations in preoperative prognostic assessment, we emphasize that management decisions should be individualized, considering both disease characteristics and patient preferences.
We believe our findings contribute valuable data to the growing body of literature on this frequently debated but clinically important topic. Our goal is to support ongoing efforts to refine risk stratification and improve patient care.
To address your comments the following changes have been made to the manuscript:
- A paragraph discussing the safety and cost-effectiveness of AS in low-risk disease through the published experience of Miyauchi et al. has been added to the introduction (lines 96-105).
- A paragraph discussing AS as a safe first-line management approach has been added to the discussion (lines 361-370)
- we expanded the paragraph discussing our previously published experience with shared physician-patient decision-making, and patients' preferences in the management of clinical solitary thyroid nodules in a global healthcare setting (lines 371-379)
- The paragraph addressing the limitations of the study has been expanded (lines 380-387)
- The conclusion has been edited (391-398)
Reviewer 3 Report
Comments and Suggestions for Authors
Dear Authors,
Thank you for addressing such a timely and highly debated topic in endocrine surgery and endocrinology—namely, the management of incidental and non-incidental papillary thyroid microcarcinoma (PTMC). Your manuscript provides valuable insight into real-world surgical practice over a ten-year period. I appreciate your contribution to this important field.
Please find below my comments and suggestions, offered with the intent to strengthen the clarity, transparency, and scientific rigor of your work.
-
Clarify the Study Aim in the Abstract and Title
The stated objective in the abstract ("Our study compares disease characteristics and outcomes of incidental vs non-incidental cases over ten years") may appear somewhat vague or incomplete. I encourage you to explicitly state why this comparison is relevant. What clinical or scientific gap were you aiming to address? Providing this rationale upfront will help readers assess the added value of reading the full manuscript. In the same vein, the title could be more specific to reflect the comparative nature and clinical relevance of your investigation. -
Inclusion and Exclusion Criteria
In lines 103–105, you mention the exclusion of patients with medullary carcinoma. However, it would be helpful to clarify whether other non-PTC histologies (e.g., follicular, poorly differentiated, or anaplastic carcinomas) were also excluded. Furthermore, did your study include both patients with a known preoperative diagnosis of PTC and those undergoing surgery for presumed benign disease (based on US and/or FNAC)? If so, explicitly stating this would reinforce the rationale for comparing truly incidental vs clinically suspected cases. -
Surgical Management and Terminology
Lines 120–121 state: “All patients underwent surgical resection, with central neck dissection for clinically or ultrasound-positive or suspicious nodes and modified neck dissection for positive lateral nodes.”
I suggest rephrasing this section to clearly distinguish between prophylactic and therapeutic central neck dissection, and to specify the criteria for lateral neck dissection. This distinction is important for understanding the extent of surgery and its alignment with current guidelines. -
Clarify Use of Acronyms
In line 122, you mention: “None of the patients opted for AS.” Please define AS as “active surveillance” at first use to avoid ambiguity. -
Pathological Review Process
Lines 123–131 describe the histopathological evaluation. It is unclear whether the external pathologist from the University of Iowa retrospectively reviewed all surgical cases from 2013 to 2022, or only those diagnosed as PTMC by local pathologists. Did this review involve full slide re-evaluation, recutting of paraffin blocks, or simply review of previously prepared sections? Was preoperative cytology also reviewed to assess diagnostic concordance? Clarifying this process is crucial, especially when incidental diagnoses are involved. A flowchart or diagram illustrating the pathological workflow would greatly enhance methodological transparency. -
Definition of “Benign” Specimens
In lines 132–135, please clarify what constitutes “benign thyroid specimens.” Was benignity determined preoperatively (e.g., based on FNAC or ultrasound) or solely confirmed postoperatively? Greater precision would help readers understand the nature of the incidental PTMCs. -
Molecular Testing
Did any patient undergo molecular testing (on cytology or histology)? Given the discussion on BRAF and TERT mutations, including this information would contextualize your findings and highlight the study’s limitations. -
Possible Typo – Line 353
In the phrase “A5 or TL are viable options,” could “A5” be a typographical error? Please verify and correct as needed. -
Sample Size of NIPTMC Group
The small number of NIPTMC patients (n = 28) is a limitation that may affect the statistical power and generalizability of group comparisons. Consider discussing this more explicitly in your limitations section. -
Lack of Active Surveillance Group
None of the patients were managed with active surveillance, which limits the ability to compare surgical outcomes with conservative management, particularly in low-risk PTMC. This may also result in overestimation of tumor aggressiveness due to the surgical selection bias. Furthermore, the predominance of total thyroidectomies, even in recent years, reflects a surgical practice not fully aligned with current de-escalation trends and guideline recommendations. -
Cultural and Logistic Influences on Management
While you mention that international patients or cultural preferences may influence treatment choices, this aspect is not formally addressed or analysed. Exploring these as potential confounding factors would strengthen your discussion. -
Shared Decision-Making and Patient Perspective
Although shared decision-making is mentioned, no structured data on patient preferences or communication quality are presented. Given the nuanced and often patient-driven nature of PTMC management, this omission is noteworthy. -
Follow-Up Data
The manuscript would benefit from more detailed reporting on follow-up, including its median duration, imaging or lab surveillance protocols, and adjuvant treatments (e.g., RAI). This would provide a clearer picture of long-term outcomes beyond the surgical phase.
I hope you find these suggestions constructive and helpful in enhancing the clarity and impact of your study. Your work contributes to the ongoing discussion about optimal PTMC management and highlights important real-world practices that merit further investigation.
Author Response
We sincerely thank you for your thoughtful and constructive comments on our manuscript. We have carefully considered each suggestion and revised the manuscript accordingly. All changes are highlighted in red. Below, we provide a detailed response to each comment.
Comment 1: Clarify the Study Aim in the Abstract and Title
The stated objective in the abstract ("Our study compares disease characteristics and outcomes of incidental vs non-incidental cases over ten years") may appear somewhat vague or incomplete. I encourage you to explicitly state why this comparison is relevant. What clinical or scientific gap were you aiming to address? Providing this rationale upfront will help readers assess the added value of reading the full manuscript. In the same vein, the title could be more specific to reflect the comparative nature and clinical relevance of your investigation.
Response 1: We agree with your suggestions. We have made changes to both the title and abstract (highlighted in red).
Comment 2: Inclusion and Exclusion Criteria
In lines 103–105, you mention the exclusion of patients with medullary carcinoma. However, it would be helpful to clarify whether other non-PTC histologies (e.g., follicular, poorly differentiated, or anaplastic carcinomas) were also excluded. Furthermore, did your study include both patients with a known preoperative diagnosis of PTC and those undergoing surgery for presumed benign disease (based on US and/or FNAC)? If so, explicitly stating this would reinforce the rationale for comparing truly incidental vs clinically suspected cases.
Response 2: Only cases with papillary thyroid carcinoma were studies. We have edited the prior sentence to read as follows: The study includes consecutive patients with papillary thyroid carcinoma over the age of 18, excluding patients undergoing combined thyroid and parathyroid surgery and patients with familial MENII syndrome." Lines 136-138
The definition of the two groups IPTMC and NIPTMC in the methods section has been edited to read was follows: "Based on our patient cohort, PTMCs were classified into two groups: 1. Incidental PTMC (IPTMC) represents tumors identified incidentally in patients referred for thyroid surgery for benign thyroid disease including multinodular goiter, Hashimoto thyroiditis, Graves’ disease, or adenoma. These tumors were predominantly identified postoperatively through pathological examination of surgical specimens. 2. Non-incidental or primary PTMC (NIPTMC) includes tumors that present clinically as thyroid nodules or abnormal lymph nodes, are diagnosed preoperatively through clinical evaluation, imaging, or fine-needle aspiration cytology (FNAC), and arise in otherwise normal thyroid tissue without associated thyroid pathology (Fig. 1)." Lines 168-175
Comment 3: Surgical Management and Terminology
Lines 120–121 state: “All patients underwent surgical resection, with central neck dissection for clinically or ultrasound-positive or suspicious nodes and modified neck dissection for positive lateral nodes.”
I suggest rephrasing this section to clearly distinguish between prophylactic and therapeutic central neck dissection, and to specify the criteria for lateral neck dissection. This distinction is important for understanding the extent of surgery and its alignment with current guidelines.
Response 3: Both central neck and lateral neck dissections were performed therapeutically. The sentence has been edited to read as follows: "All patients underwent surgical resection, with therapeutic central neck dissection for clinically or ultrasound-positive or suspicious nodes and modified neck dissection for positive lateral nodes." Lines 152-153
Comment 4: Clarify Use of Acronyms
In line 122, you mention: “None of the patients opted for AS.” Please define AS as “active surveillance” at first use to avoid ambiguity.
Response 4: This has been corrected. Line 153-154
Comment 5: Pathological Review Process
Lines 123–131 describe the histopathological evaluation. It is unclear whether the external pathologist from the University of Iowa retrospectively reviewed all surgical cases from 2013 to 2022, or only those diagnosed as PTMC by local pathologists. Did this review involve full slide re-evaluation, recutting of paraffin blocks, or simply review of previously prepared sections? Was preoperative cytology also reviewed to assess diagnostic concordance? Clarifying this process is crucial, especially when incidental diagnoses are involved. A flowchart or diagram illustrating the pathological workflow would greatly enhance methodological transparency.
Response 5: The histopathological review process description has been modified to include further details as follows: "Histopathological examination was performed by two board-certified pathologists experienced in thyroid pathology from Jordan Hospital (Amman, Jordan). Thyroid specimens resected for benign thyroid disease were thoroughly examined for incidental microcarcinomas (≤1 cm). Over a one-month period, the first author, a surgical pathologist and cytopathologist from the University of Iowa (Iowa, USA) performed a blinded re-evaluation of all original slides of PTMC surgical cases diagnosed by local pathologists during the period of the study (2013-2022) and classified all PTMCs based on the updated 2022 WHO classification of thyroid neoplasms (13). Pathologic examination of tumors included assessments of histologic subtype, size, site, margins, bilaterality, multifocality, presence or absence of extrathyroidal extension (ETE), and lymphovascular invasion."Lines 155-163
FNAC cases were available for review, but not reviewed.
Comment 6: Definition of “Benign” Specimens
In lines 132–135, please clarify what constitutes “benign thyroid specimens.” Was benignity determined preoperatively (e.g., based on FNAC or ultrasound) or solely confirmed postoperatively? Greater precision would help readers understand the nature of the incidental PTMCs.
Response 6: This has been corrected. These specimens represent thyroidectomy specimens resection for benign thyroid disease, multi nodular goiter, Hashimoto thyroiditis and adenoma. This has been edited to read as follows: "Thyroid specimens resected for benign thyroid disease were thoroughly examined for incidental microcarcinomas (≤1 cm)." Lines 156-157
Comment 7: Molecular Testing
Did any patient undergo molecular testing (on cytology or histology)? Given the discussion on BRAF and TERT mutations, including this information would contextualize your findings and highlight the study’s limitations.
Response 7: None of the patients underwent molecular testing as these tests are not readily available for routine clinical practice in Jordan due to the prohibitive cost and lack of insurance. This has been added to the end of the paragraph discussing BRAF and TERT. "None of our cases underwent molecular testing, as these tests are not readily available for routine clinical use due to the prohibitive cost and lack of insurance coverage." Lines 310-311
Comment 8: Possible Typo – Line 353
In the phrase “A5 or TL are viable options,” could “A5” be a typographical error? Please verify and correct as needed.
Response 8: This is a typographical error. It should read AS. This has been corrected
Comment 9: Sample Size of NIPTMC Group
The small number of NIPTMC patients (n = 28) is a limitation that may affect the statistical power and generalizability of group comparisons. Consider discussing this more explicitly in your limitations section.
Response 9: The limitations paragraph has been edited and expanded as follows:
"This study has several limitations. It is a retrospective analysis from a single institution. The relatively small sample size of the NIPTMC group (n=28) represents a limitation that may impact the statistical power and reduce generalizability of our findings. None of the patients underwent AS; therefore, disease progression data is unavailable. Furthermore, the lack of an AS cohort limits the ability to compare outcomes with more conservative treatment strategies. Molecular testing was not performed on any of our cases. Although molecular testing can assist in identifying high risk mutations, it is not currently accessible for routine clinical use in Jordan. The high cost of molecular testing continues to pose a significant financial barrier to widespread use, particularly in resource-constrained settings." Lines 380-387
Comment 10: Lack of Active Surveillance Group
None of the patients were managed with active surveillance, which limits the ability to compare surgical outcomes with conservative management, particularly in low-risk PTMC. This may also result in overestimation of tumor aggressiveness due to the surgical selection bias. Furthermore, the predominance of total thyroidectomies, even in recent years, reflects a surgical practice not fully aligned with current de-escalation trends and guideline recommendations.
Response 10: We recognize this as a limitation of this study. In addition to highlighting this in the limitations paragraph (please see response 9), we have made modifications to the discussion (lines 361-380), as follows:
Miyauchi et al. (9) have reported that based on the accumulation of long-term data, AS for low-risk PTMC is a safe management strategy and offers greater benefits than immediate surgery, for both patients and society. Unfortunately, none of our patients opted for AS. In our experience, cultural perceptions and heightened anxiety surrounding cancer, contribute to a preference for definitive surgical management, potentially limiting the adoption of more conservative approaches. Additionally, the success of active surveillance relies heavily on patient adherence to follow-up and recurrence monitoring protocols. However, concerns regarding long-term compliance may deter some patients from opting for this management strategy. Ultimately, increasing awareness and education among endocrinologists, surgeons, patients and the public is essential to promote AS as a safe and effective management option for low-risk disease.
As previously reported by Bashir et al. (50), our experience shows patients favor TT due to fear of cancer recurrence or potential future surgeries and for ease of follow-up especially for international patients seeking treatment. Preoperatively, the fear of malignancy was the reason for choosing surgery in 35.1% of patients, avoiding FNAC in 20.2% of patients and refusal of AS. Patient preference-based decisions for TT vs. TL were less guideline-concordant than decisions based on frozen section (21% vs. 79%), with more TTs performed (74.4% vs. 41%, P = 0.001). Notably, in a global healthcare setting, molecular testing is not readily available due to cost and lack of insurance coverage. Improving patient understanding of the disease and up-to-date management options is crucial for more effective shared decision-making (51). The ATA risk stratification for PTMC aids clinical decision-making (4)
Comment 11: Cultural and Logistic Influences on Management
While you mention that international patients or cultural preferences may influence treatment choices, this aspect is not formally addressed or analysed. Exploring these as potential confounding factors would strengthen your discussion.
Response 11: Generally international patients are seeking treatment without delays. The number of national vs international patients for IPTMC and NIPTMC is listed in table 3. We added a paragraph referencing the previously published study by corresponding author and members of the group (lines 371-380).
Bashir AY, El-Zaheri MM, Obed AH, Abufares F, Haddadin M, Annab HZ, Abu-Hijleh MO, Bashir MA, Bashir AA. Patients’ preferences impact on decision-making for clinical solitary thyroid nodule in a global healthcare setting: a clinical study. s.l. : Series Endo Diab Met, 2021. 3(2):48-58. As follows:
"As previously reported by Bashir et al. (50), our experience shows patients favor TT due to fear of cancer recurrence or potential future surgeries and for ease of follow-up especially for international patients seeking treatment with minimal delays. Preoperatively, the fear of malignancy was the reason for choosing surgery in 35.1% of patients, avoiding FNAC in 20.2% of patients and refusal of AS. Patient preference-based decisions for TT vs. TL were less guideline-concordant than decisions based on frozen section (21% vs. 79%), with more TTs performed (74.4% vs. 41%, P = 0.001). Notably, in a global healthcare setting, molecular testing is not readily available due to cost and lack of insurance coverage. Improving patient understanding of the disease and up-to-date management options is crucial for more effective shared decision-making (51). The ATA risk stratification for PTMC aids clinical decision-making (4)."
Comment 12: Shared Decision-Making and Patient Perspective
Although shared decision-making is mentioned, no structured data on patient preferences or communication quality are presented. Given the nuanced and often patient-driven nature of PTMC management, this omission is noteworthy.
Response 12: A survey to obtain comprehensive information about patient preferences is performed followed by a face-to face discussion with the provider. These discussion can happen over more than one visit as testing results are obtained. We have included a paragraph discussing the findings from a previously published study by the corresponding author and members of the group around patient preferences and the management of solitary thyroid nodules (lines 371-380). Please refer to response 11.
Comment 13: Follow-Up Data
The manuscript would benefit from more detailed reporting on follow-up, including its median duration, imaging or lab surveillance protocols, and adjuvant treatments (e.g., RAI). This would provide a clearer picture of long-term outcomes beyond the surgical phase.
Response 13: We agree and have added a paragraph discussing the follow duration and protocol description in the Materials and Methods portion of the manuscript (lines 164-167) as follows:
"The median patient follow-up duration was five years. The recurrence surveillance protocol included clinical examination, thyroid ultrasound and laboratory testing for thyroglobulin, antithyroglobulin antibodies and TSH every three months for the first year, then every sixth months for two additional years and once a year thereafter."
Round 2
Reviewer 3 Report
Comments and Suggestions for Authors
Dear Authors,
Thank you very much for your detailed responses and for having clarified the points I previously raised. I truly appreciate your effort in addressing my questions.
I just have one final comment. Considering the very limited number of lobectomies reported in your series — and, if I am not mistaken, the fact that none were performed when a preoperative diagnosis of papillary microcarcinoma (PTMC) was available (as shown in Table 3, the 12 lobectomies appear to have occurred only in incidental cases) — could you further characterize these conservative cases?
More specifically:
What clinical or surgical reasoning led to the choice of lobectomy in those incidental PTMC cases?
Why was a conservative approach never adopted when the PTMC diagnosis was already known preoperatively?
Additionally, do you include in the "total thyroidectomy" group also those cases in which completion thyroidectomy was performed after the histological diagnosis?
Thank you once again for your thoughtful work.
Author Response
We thank you for your thoughtful comments and constructive suggestions, which have helped us improve the quality and clarity of our manuscript. Below, we provide responses to your questions.
Question:
Considering the very limited number of lobectomies reported in your series — and, if I am not mistaken, the fact that none were performed when a preoperative diagnosis of papillary microcarcinoma (PTMC) was available (as shown in Table 3, the 12 lobectomies appear to have occurred only in incidental cases) — could you further characterize these conservative cases?
More specifically:
What clinical or surgical reasoning led to the choice of lobectomy in those incidental PTMC cases?
Response:
As shown in tables 3, twelve lobectomies were performed in incidental cases. These included eight patients with multinodular goiter, three with Hashimoto thyroiditis and one with a follicular adenoma. Upon histopathologic evaluation, all tumors were classified as low-risk. Accordingly, following patient counseling and a shared decision-making process, the patients endorsed the conservative management approach.
A paragraph to that effect has been incorporated into the revised manuscript (lines 238-242, highlighted).
Question:
Why was a conservative approach never adopted when the PTMC diagnosis was already known preoperatively?
Response:
As mentioned in the Materials and Methods section (lines 150-151), “Informed consent and patient counseling covered treatment plans, intraoperative decisions, and patient preferences.” Based on our experience patients favor total thyroidectomy primarily due to fear of cancer recurrence or potential future surgery. The discussion refers to our previously published work, reference number 50: Bashir AY, El-Zaheri MM, Obed AH, Abufares F, Haddadin M, Annab HZ, Abu-Hijleh MO, Bashir MA, Bashir AA. Patients’ preferences impact on decision-making for clinical solitary thyroid nodule in a global healthcare setting: a clinical study. s.l. : Series Endo Diab Met, 2021. 3(2):48-58, in lines 376-384 as follows:
“As previously reported by Bashir et al. (50), our experience shows patients favor TT due to fear of cancer recurrence or potential future surgeries and for ease of follow-up especially for international patients seeking treatment with minimal delays. Preoperatively, the fear of malignancy was the reason for choosing surgery in 35.1% of patients, avoiding FNAC in 20.2% of patients and refusal of AS. Patient preference-based decisions for TT vs. TL were less guideline-concordant than decisions based on frozen section (21% vs. 79%), with more TTs performed (74.4% vs. 41%, P = 0.001). Notably, in a global healthcare setting, molecular testing is not readily available due to cost and lack of insurance coverage. Improving patient understanding of the disease and up-to-date management options is crucial for more effective shared decision-making (51). The ATA risk stratification for PTMC aids clinical decision-making (4).”
Patients’ preference for definitive surgery vs AS was also discussed in the manuscript (lines 368-375) as follows:
“In our experience, cultural perceptions and heightened anxiety surrounding cancer, contribute to a preference for definitive surgical management, potentially limiting the adoption of more conservative approaches. Additionally, the success of active surveillance relies heavily on patient adherence to follow-up and recurrence monitoring protocols. However, concerns regarding long-term compliance may deter some patients from opting for this management strategy. Ultimately, increasing awareness and education among endocrinologists, surgeons, patients and the public is essential to promote AS as a safe and effective management option for low-risk disease.”
The manuscript outlines the management strategy at Jordan Hospital in lines 352-365, as follows:
“The management strategy at Jordan Hospital Medical Center includes:
- High-Risk PTMCs: PTMCs with high-risk features such as ETE, lymph node metastasis, aggressive histologic subtype, and proximity to the trachea or recurrent laryngeal nerve require aggressive surgical management (49), including total thyroidectomy, lymph node dissection for clinical or US-positive nodes, and radioactive iodine ablation.
- Benign Thyroid Disease: Patients with presumed benign thyroid disease are referred for surgery based on specific indications related to the thyroid pathology. Total thyroidectomy is often required for associated conditions like MNG, toxic nodular goiter, Graves’ disease, and Hashimoto thyroiditis (HT). PTMC diagnosis in these cases is typically made postoperatively through pathological examination of thyroidectomy specimens.
- Low-Risk PTMCs: For preoperatively diagnosed low-risk PTMCs, including unifocal intrathyroidal tumors with clinically negative nodes, management is determined through a physician-patient shared decision-making process, considering clinical evidence, patient preferences, beliefs, comorbidities, and available resources (50,51,45). AS or TL are viable options.”
Question:
Additionally, do you include in the "total thyroidectomy" group also those cases in which completion thyroidectomy was performed after the histological diagnosis?
Response:
Yes, completion thyroidectomies are included in the total thyroidectomy group as stated in the manuscript as follows (lines 235-237):
“Total thyroidectomy was performed in all cases of NIPTMCs and MF-PTMCs (100%), compared to 88% in IPTMCs and unifocal PTMCs (P < 0.001) (Table 4). This included eight completion thyroidectomies requested by female patients aged 32-36 years due to concerns about malignancy risk.”